# FLEXIQuant-LF to quantify protein modification extent in label-free proteomics data

Christoph N Schlaffner[1,2†], Konstantin Kahnert[3,4,5†], Jan Muntel[3,6†‡], Ruchi Chauhan[1§], Bernhard Y Renard[4,7], Judith A Steen[1,2], Hanno Steen[3,6,8*]

[1]F.M. Kirby Neurobiology Center, Boston Children's Hospital, Boston, United States; [2]Department of Neurology, Harvard Medical School, Boston, United States; [3]Department of Pathology, Boston Children's Hospital, Boston, United States; [4]Bioinformatics Unit (MF1), Robert Koch Institute, Berlin, Germany; [5]Department of Medical Biotechnology, Institute of Biotechnology, Technische Universität Berlin, Berlin, Germany; [6]Department of Pathology, Harvard Medical School, Boston, United States; [7]Data Analytics and Computational Statistics, Hasso-Plattner-Institute, Faculty of Digital Engineering, University of Potsdam, Potsdam, Germany; [8]Precision Vaccines Program, Boston Children's Hospital, Boston, United States

*For correspondence:
Hanno.Steen@childrens.harvard.
edu

†These authors contributed
equally to this work

Present address: ‡Biognosys
AG, Schlieren, Switzerland;
§Grousbeck Gene Therapy
Center, Gene Transfer Vector
Core, Massachusetts Eye and Ear
Infirmary/Harvard Medical
School, Boston, United States

Competing interests: The
authors declare that no
competing interests exist.

Reviewing editor: Akhilesh
Pandey, Mayo Clinic, United
States

**Abstract** Improvements in LC-MS/MS methods and technology have enabled the identification of thousands of modified peptides in a single experiment. However, protein regulation by post-translational modifications (PTMs) is not binary, making methods to quantify the modification extent crucial to understanding the role of PTMs. Here, we introduce FLEXIQuant-LF, a software tool for large-scale identification of differentially modified peptides and quantification of their modification extent without knowledge of the types of modifications involved. We developed FLEXIQuant-LF using label-free quantification of unmodified peptides and robust linear regression to quantify the modification extent of peptides. As proof of concept, we applied FLEXIQuant-LF to data-independent-acquisition (DIA) data of the anaphase promoting complex/cyclosome (APC/C) during mitosis. The unbiased FLEXIQuant-LF approach to assess the modification extent in quantitative proteomics data provides a better understanding of the function and regulation of PTMs. The software is available at https://github.com/SteenOmicsLab/FLEXIQuantLF.

## Introduction

Most cellular processes are regulated by post-translational modifications (PTMs). Given the sensitivity of current mass spectrometric analyses, even non-functional basal PTMs can be identified by mass spectrometry. Thus, an understanding of the stoichiometry and changes thereof are crucial to understanding function. Numerous bioinformatics tools that enable unbiased and large-scale identification of modified peptides and localization of the modification site such as MSFragger (*Kong et al., 2017*), MetaMorpheus (*Solntsev et al., 2018*), or TagGraph (*Devabhaktuni et al., 2019*) have been developed. With steadily increasing qualitative information about the identity of PTMs (*Grimsrud et al., 2010*; *Choudhary et al., 2014*; *Kim et al., 2011*; *Melo-Braga et al., 2014*; *Udeshi et al., 2013*), tools to investigate the modification quantity, that is the proportion of the protein carrying PTMs, become increasingly important. Thus far, analysis of the PTM extent has mainly focused on quantifying the modified peptides, for example using PTM enrichment methods (*Pinkse et al., 2004*; *Rush et al., 2005*; *Zhou et al., 2011*; *Lundby et al., 2012*; *Lundby et al., 2013*), heavy isotope-labeled synthetic peptides (*Gerber et al., 2003*) or by enzymatically removing

(*Yamagata et al., 2002*; *Wu et al., 2011*) PTMs. These approaches require prior knowledge of the PTMs and have been primarily applied to the phosphoproteome.

The development of FLEXIQuant (*F*ull-*L*ength-*E*xpressed Stable *I*sotope-labeled Proteins for *Quant*ification) (*Singh et al., 2009*; *Singh et al., 2012a*), for the first time, enabled the quantification of the extent of modification across a whole protein without prior knowledge of modification identities using an indirect approach of quantifying only the unmodified peptides. FLEXIQuant relies on the principle that the total number of molecules of a given protein is conserved in a sample, thus the number of molecules of each unique peptide derived from that protein will be equal. If a peptide is modified chemically by a PTM this would reduce the abundance of its unmodified cognate in the peptide pool. In FLEXIQuant, this idea is realized by analyzing the peptide pool of the sample with an added unmodified heavy isotope-labeled full-length reference protein. The extent of modification is calculated by comparing the quantities of labeled reference peptides derived from the standard to the unlabeled cognate peptides from the sample, that is the unlabeled endogenous protein. The advantage of this approach is the precise and accurate quantification of the degree of abundance reduction due to a modification as defined by *Wold, 1981* or amino acid substitutions of each quantified unmodified peptide and the ability to calculate the absolute concentration. FLEXIQuant and derivatives thereof have been applied to study for example Tiki1 modification in head formation in frogs (*Zhang et al., 2012*), to gain insights into the phosphorylation dynamics of GSK3β-dependent phosphorylation of DCX (*Singh et al., 2012b*), the cell-cycle-dependent phosphorylation of KifC1 (*Singh et al., 2014*) or to investigate PTM of Tau in Alzheimer's disease (*Mair et al., 2016*). However, the limitation of this method is the requirement of a purified, unmodified, labeled reference protein and thus, it is labor intensive and has largely been used to investigate individual proteins. Recently, a PTM analysis pipeline without the need for an internal reference protein has been published (*Bagwan et al., 2018*). Although this is encouraging, the quantification relies on isobaric labeling approaches which are expensive to conduct due to the high cost of the labeling reagents and is therefore not widely applicable, especially not for large-scale experiments. This highlights the need for bioinformatic tools suitable for the analysis of label-free experiments.

Here, we introduce FLEXIQuant-LF as an unbiased, label-free computational tool to indirectly detect modified peptides and to quantify the degree of modification based solely on the unmodified peptide species building upon the FLEXIQuant (*Singh et al., 2009*) idea developed in our lab. We developed this approach to identify and elucidate differential protein modification extent in commonly analyzed time series or case-control studies. In these studies, a timepoint or control group is used as reference to enable a FLEXIQuant-like modification analysis. The key requirement of FLEXIQuant-LF is the unbiased, comprehensive and highly reproducible quantification of peptides, therefore, we use a data-independent acquisition (DIA) strategy (namely, SWATH [*Gillet et al., 2012*]). Here, we demonstrate the utility of FLEXIQuant-LF by applying our method to interrogate the peptide-resolved modification dynamics of the anaphase-promoting complex/cyclosome (APC/C) during mitosis (*Singh et al., 2009*; *Steen et al., 2008*).

# Materials and methods

**Key resources table**

| Reagent type (species) or resource | Designation | Source or reference | Identifiers | Additional information |
|---|---|---|---|---|
| Cell line (*Homo sapiens*) | HeLa S3 | ATCC | Cat# CCL-2.2, RRID:CVCL_0058 | |
| Antibody | Mouse monoclonal CDC27 antibody (AF3.1) | Santa Cruz | Cat# sc-9972 | IP (1:3) |
| Antibody | Mouse polyclonal Normal mouse IgG | Santa Cruz | Cat# sc-2025 | IP (1:6) |
| Peptide, recombinant protein | HRM calibration peptides (iRT peptides) | Biognosys | Cat# Ki-3002 | |
| Peptide, Recombinant protein | Trypsin (sequencing grade modified trypsin) | Promega | V517 | |

*Continued on next page*

*Continued*

| Reagent type (species) or resource | Designation | Source or reference | Identifiers | Additional information |
|---|---|---|---|---|
| Chemical compound, drug | Nacodazole | Sigma Aldrich | Cat# M1404 | 100 ng/ml |
| Chemical compound, drug | Penicillin and streptomycin mix | Invitrogen | Cat# 15140–122 | 100 µg/ml |
| Chemical compound, drug | Halt Protease and Phosphatase Inhibitor | Thermo Fisher | Cat# 78442 | |
| Software, algorithm | FLEXIQuant-LF | This paper | | See Abstract, Materials and methods, Data and Software Availability |
| Software, algorithm | MaxQuant v1.5.2.8 | Cox lab, Max Planck Institute of Biochemistry | https://www.maxquant.org | |
| Software, algorithm | ProteinPilot v4.5.1 | Sciex | https://sciex.com/ | |
| Software, algorithm | Spectronaut v7.0 | Biognosys | https://biognosys.com | |
| Other | DMEM media | Invitrogen | Cat# 11965 | |
| Other | FBS | Invitrogen | Cat# 26140–079 | |
| Other | L-glutamine | Invitrogen | | ThermoFisher: Cat# 25030149 |
| Other | 4–12% SDS-PAGE gel | Invitrogen | Cat# NP0329BOX | |
| Other | MES buffer | Invitrogen | Cat# NP0002-02 | |
| Other | Thymidine | Sigma Aldrich | Cat# T1895 | |
| Other | RIPA Lysis Buffer | Santa Cruz | Cat# sc-24948 | |
| Other | Affi-Prep Protein A beads | Biorad | Cat# 1560006 | |
| Other | Formic acid (FA) | Thermo Fisher | Cat# A117-50 | |
| Other | Acetonitrile (ACN) | Thermo fisher | Cat# A955-4 | |
| Other | Water | Thermo fisher | Cat# W6-4 | |
| Other | TripleTOF 5600 | Sciex | https://sciex.com/ | |
| Other | Nano cHiPLC trap column (200 µm x 0.5 mm Reprosil C18 3 µm 120 Å) | Eksigent | Cat# 804–00016 | |
| Other | Nano cHiPLC column (75 µm x 15 cm Reprosil C18 3 µm 120 Å) | Eksigent | Cat# 804–00011 | |
| Other | DTT | Sigma Aldrich | Cat# D9779 | |

## FLEXIQuant-LF algorithm and benchmarking

FLEXIQuant-LF indirectly identifies modified peptides by means of pinpointing unmodified peptides whose intensities strongly deviate from those of a reference sample using robust linear regression. Based on the distance of each peptide to the regression line, Relative Modification (RM) scores are calculated that, in analogy to the light to heavy ratio of the original FLEXIQuant method, correspond to the degree of modification (*Figure 1B*).

More specifically, the FLEXIQuant-LF algorithm trains a linear regression model for each sample in the input file using the random sample consensus (RANSAC) algorithm (*Fischler and Bolles, 1981*) to identify outliers iteratively and fit the model only based on the inliers, that is based on stably and consistently quantified unmodified peptides. For this, peptide intensities of each sample, as defined by the user as a single row in the input data file, are used as dependent variable ($y_i$), peptide intensities of a user defined reference sample as independent variable ($x_i$) and no intercept is fitted (*Figure 1B2*):

$$y(i) = \beta x_i + \epsilon \tag{1}$$

In Formula 1, the slope of the regression line is described by $\beta$, whereas $\epsilon$ represents an error

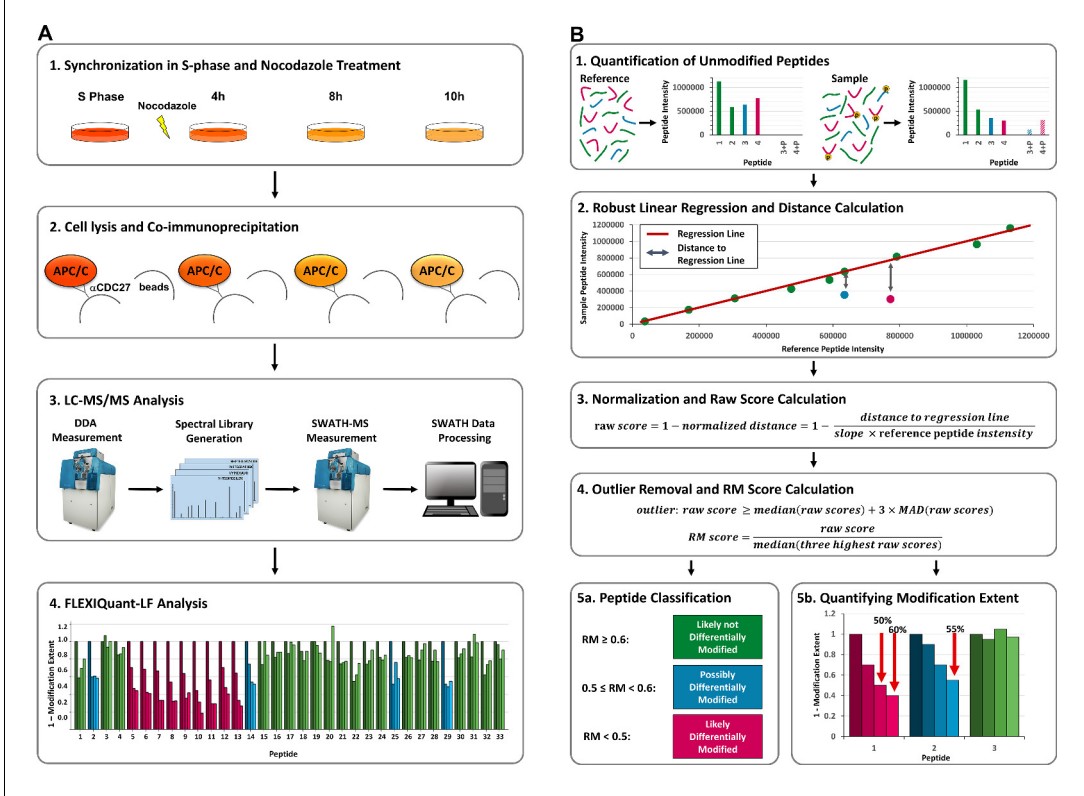

**Figure 1.** Workflow and FLEXIQuant-LF concept. (**A**) Workflow. (**A1**) HeLa cells were synchronized in S-phase using thymidine (dark orange). Upon release from thymidine block, cells were treated with nocodazole and samples were collected after 4 hr (medium dark orange), 8 hr (orange), and 10 hr (light orange). (**A2**) APC/C was co-immunoprecipitated using an anti-CDC27 antibody. (**A3**) All samples were trypsinized separately and analyzed by LC-MS/MS in DDA mode to generate a spectral library, which was subsequently analyzed by SWATH-MS (see ***Supplementary file 1*** for raw peptide intensities of all quantified APC/C proteins). (**A4**) FLEXIQuant-LF-based differential modification analysis of APC/C proteins (see ***Supplementary file 2*** for all resulting RM scores). (**B**) FLEXIQuant-LF overview. (**B1**) Firstly, intensities of unmodified peptides are used to indirectly identify and quantify the modification extent of a protein using the following steps. (**B2**) A RANSAC-based robust linear regression model is fitted to the intensities of unmodified peptide species using a reference sample as independent variable and the sample of interest as dependent variable and the vertical distance to the regression line of each peptide is determined. (**B3**) For each peptide, the distance is then normalized by dividing by the slope of the regression line multiplied by the intensity of the peptide in the reference sample and the result is subtracted from one to yield raw scores. (**B4**) Peptides with a raw score above three standard deviations of the median (MAD) of all raw scores are classified as outliers and excluded from the subsequent RM score calculation. The remaining raw scores are then scaled using the median of the three highest raw scores resulting in a metric, termed RM score which is equal to one minus the extent of modification. (**B5**) Lastly, peptides are classified in three categories based on their RM scores and the extent of modification is visualized: (i) RM score <0.5: peptide is likely differentially modified (magenta bars), (ii) 0.5 ≤ RM score<0.6: peptide is possibly differentially modified (blue bars), and (iii) RM score ≥0.6: peptide is likely not differentially modified (green bars).

term. To increase reproducibility and enhance the fit of the linear regression model to the data the algorithm is executed 30 times and the best model, based on $r^2$ scores, is selected. Subsequently, vertical distances between each peptide intensity and the obtained regression line are determined, that is intensity differences between measured and expected peptide intensities. To correct for sample-specific differences in individual protein abundance and peptide-specific differences in intensity, distances are normalized by dividing them by the expected intensity, that is the reference intensity of the peptide from the independent variable multiplied by the slope of the regression line. FLEXIQuant-LF raw scores are then calculated by subtracting the normalized distances from 1 (***Figure 1B3***). For each sample, peptides with a raw score above three standard deviations from the median (MAD) of all raw scores are classified as outliers and excluded from the subsequent RM score (Relative Modification score) calculation. RM scores are calculated by dividing each inlier raw score by the median of the three highest raw scores (after removing outliers) for each sample, corresponding to one minus the extent of modification as defined by FLEXIQuant (***Figure 1B4***). A detailed derivation of the method can be found in our repository on GitHub: https://github.com/SteenOmicsLab/

FLEXIQuantLF (copy archived at swh:1:rev:4ea3945f86ba477227c89e9ced75fc23751355ac; *Kahnert and Steen Lab, 2020*).

The FLEXIQuant-LF method was implemented in Python 3.7.3 using Scikit-learn 0.21.2 (*Pedregosa, 2011*), NumPy 1.16.4 (*van der Walt et al., 2011*), Pandas 0.24.2 (*McKinney, 2010*), SciPy 1.2.1 (*Jones, 2001*), Matplotlib 3.1.0 (*Hunter, 2007*), Seaborn 0.9.0 (*Waskom, 2017*). Parameters of the RANSAC algorithm are set as FLEXIQuant-LF defaults as follows: max_trials = 1000, base_estimator = linear_model.LinearRegression, min_samples = 0.5, stop_probability = 1, loss='squared_loss', residual_threshold = MAD(sample intensities)$^2$. The Fit_intercept parameter of the linear regression algorithm is set to False. The graphical user interface was built using QT Designer 5.13.0 (*Espoo, 2019*), QT 5.9.7 (*The QT Company, 2019*), and PyQT 5.13.2 (*Riverbank Computing Limited, 2019*). PyInstaller 3.6 (*PyInstaller Development Team, 2019*) was used to create the executable file. The command line interface version was built utilizing Click 7.0 (*The Pallets Projects, 2020*).

To benchmark FLEXIQuant-LF and identify differentially modified peptides in the time course experiment, peptide intensities of the proteins APC1, APC4, APC5, APC7, APC10, APC15, APC16, CDC16, CDC20, CDC23, CDC26, and CDC27 from the well-studied APC/C were extracted. Peptides used for quantification had to be unmodified, or only modified with oxidation on methionine and/or carbamidomethylation on cysteine residues. Furthermore, they had to be quantifiable in all LC/MS runs. Median intensities of all replicates of each time point were calculated. Time point 0 (S phase) was defined as reference time point for FLEXIQuant-LF application using the default values described above. Peptides were then classified in three categories based on their RM scores at time point 10 hr: (1) RM score <0.5: peptide is likely differentially modified (magenta bars), (2) 0.5 $\leq$ RM score<0.6: peptide is possibly differentially modified (blue bars), and (3) RM score $\geq$0.6: peptide is likely *not* differentially modified (green bars) (*Figure 1B5*).

## Reproducibility and quality assessment

The reproducibility of the developed approach was tested using a large, inhouse MRM data set consisting of 81 samples of which 26 were reference group samples. We ran FLEXIQuant-LF 1000 times on the same protein and analyzed the resulting slopes to determine the number and frequency of different outcomes. To further test the quality of the classification and quantification accuracy and independent publicly available, DIA dataset was downloaded from the PRIDE repository (PXD005573) (*Bruderer et al., 2017*). Search results (file 'R Fig1 HeLa 30kMS1_Report.xls') were filtered to remove all proteins that were identified with any modification. Additionally, only peptides uniquely matching a single protein were kept. To establish a ground truth, one-third of the remaining proteins were chosen randomly for further analysis. For each protein, four peptides were selected at random and a reduction factor between 0.0 and 1.0 was generated randomly from a uniform distribution to reduce the intensity of each of these peptides for each of the three replicates. For both, the unchanged and >1400 in silico modified peptides, FLEXIQuant-LF was applied with default parameters. Additionally, as parameter for the reference group all three replicates were selected. Resulting RM scores of in silico modified peptides were compared to the RM scores of unchanged peptides and the RM score difference calculated as percentage of change between the two. The resulting values were then correlated with the expected change factors. Sensitivity and precision were also evaluated for the three classification categories. Furthermore, total RM score errors as defined by the difference between the expected change factor and the calculated RM score changes were compared to the number of peptides measured per protein to assess robustness of the FLEXIQuant-LF approach.

## Application improvement using a 'superprotein' approach

FLEXIQuant-LF results improve the more peptides are available to determine the linear regression. Two proteins (APC15, APC16) of APC/C are small and thus resulted in fewer than five quantified peptides in our experiments. To extend our analysis of APC/C to these proteins with an insufficient number of quantified peptides, we approached APC/C as a 'superprotein' drawing on the well-studied equal stoichiometry of APC/C components in the complex (with the exception of CDC20, which was omitted from this 'superprotein' approach, as it is known to change in abundance during M-phase (see results)). Therefore, all reliably quantified peptides were analyzed together as if

they were derived from a pan-APC/C 'superprotein' following the steps described above for single proteins up to raw score calculation. Peptides were then split according to the proteins they originate from and RM scores were subsequently calculated for each protein separately to improve quantification accuracy.

## Cell line

STR authenticated HeLa S3 cells were purchased from ATCC and assayed negative for mycoplasma contamination. HeLa S3 cells were grown in DMEM media (Invitrogen, Carlsbad, CA) supplemented with 10% FBS (Invitrogen), 2 mM L-glutamine (Invitrogen), 100 µg/ml penicillin and streptomycin mix (Invitrogen).

## Sample generation and co-IP

For S phase, cells were treated with 2 mM thymidine (Sigma, St. Louis, MO) at 80% confluency for 20 hr and cultured for 8 hr in media without thymidine. For M phase, thymidine-treated cells were cultured for 3 hr in fresh media and treated with 100 ng/ml nocodazole (Sigma) and were collected at 4 hr, 8 hr, and 10 hr time points. Cells were washed with PBS and lysed in 1x RIPA lysis buffer (Santa Cruz, Dallas, TX) supplemented with 1 M DTT (Sigma), 10% glycerol, protease and phosphatase inhibitors (HALT, Thermo Fisher Scientific, Waltham, MA) using a bead beater homogenizer (Precellys, Rockville, MD). Cell lysate volumes of 300 µl with 2–3 mg protein were precleared with 20 µl Affi-Prep Protein A beads (Biorad, Hercules, CA) and 20 µg mouse IgG (Santa Cruz) for 2 hr at 4°C with end-on-end rotation. Affi-Prep Protein A beads were washed twice with six volumes of buffer A (10 mM phosphate buffer, pH 7.4, 120 mM NaCl, 2.7 mM KCl, 0.1% Triton-X), twice with six volumes of buffer B (10 mM phosphate buffer, pH 7.4, 120 mM NaCl, 300 mM KCl, 0.1% Triton-X) and then once more with six volumes of buffer A. Freshly prepared beads (20 µl) conjugated with 20 µg CDC27 antibody (AF3.1, Santa Cruz) were incubated with precleared lysates for 2 hr at 4°C with end-over-end rotation. As control, the experiment was repeated using beads without conjugated antibody (controls). The beads were washed as described above. The immunoprecipitate was eluted from beads by boiling in 1x Laemmli buffer with β-mercaptoethanol.

## Sample preparation/digestion

The samples were separated on a 4–12% SDS-PAGE gel (Invitrogen) in 1x MES buffer (Invitrogen) and the excised band was subjected to in-gel digestion. In brief, proteins were reduced with DTT, alkylated with iodoacetamide and digested with trypsin. After extraction of the peptides with acetonitrile, peptides were dried down and resuspended in a buffer containing 5% formic acid/5% acetonitrile/90% water. HRM calibration peptides (Biognosys, Schlieren, Switzerland) were added to the samples prior to analysis according to manufacturer instructions.

## Spectral library generation

To generate a spectral library, we analyzed each sample once using a standard data-dependent acquisition (DDA) method on a TripleTOF 5600 mass spectrometer (Sciex, Framingham, MA) coupled online to a nanoLC system (Sciex/Eksigent, Dublin, CA), equipped with an LC-chip system (cHiPLC nanoflex, Eksigent, trapping column: Nano cHiPLC trap column 200 µm x 0.5 mm Reprosil C18 3 µm 120 Å, analytical column: Nano cHiPLC column 75 µm x 15 cm Reprosil C18 3 µm 120 Å). Peptides were separated by a linear gradient from 95% buffer A (0.2% FA in water)/5% buffer B (0.2% FA in ACN) to 70% buffer A/30% buffer B within 90 min. The mass spectrometer was operated in data-dependent TOP50 mode with the following settings: MS1 mass range 300–1700 Th with 250 ms accumulation time; MS2 mass range 100–1700 Th with 50 ms accumulation time and following MS2 selection criteria: UNIT resolution, intensity threshold 100 cts; charge states 2–5. Dynamic exclusion was set to 17 s.

The DDA data were searched with MaxQuant (v1.5.2.8) (*Cox and Mann, 2008*) against the human UNIPROT database (only reviewed entries, downloaded on October 31, 2014) using the .WIFF files without additional file conversion. The protein sequence database was appended with common laboratory contaminants (cRAP, version 2012.01.01) and the iRT fusion protein sequence (*Escher et al., 2012*) (Biognosys) resulting in 20,296 entries. The following settings were applied: trypsin with up to two missed cleavages; mass tolerances set to 0.1 Da for the first search and 0.01 for the main search.

Oxidation of M (+15.995 Da), phosphorylation of S, T, Y (+79.966 Da) and acetylation of N-TERM (+42.011 Da) were selected as dynamic modifications and carbamidomethylation of C (+57.021 Da) was selected as a static modification. FDR was set to 1% at both the peptide and protein level. The default settings were used for all other search parameters. Finally, a spectral library based on the MaxQuant search results was generated in Spectronaut 7.0 (Biognosys) using the following settings: Q value cut-off of 0.01, and a minimum of 3 and a maximum of 6 fragment ions.

To improve PTM identification output, the DDA data were also searched in Protein Pilot (v4.5.1, Sciex) using the same database described above and the following settings: sample type – identification, cys alkylation – iodoacetamide, digestion – trypsin, instrument – TripleTOF 5600, special factors – phosphorylation emphasis and gel-based ID, ID focus – biological modifications, search effort – thorough ID. For the ProteinPilot searches, it is not necessary to select the instrument mass accuracy, number of missed cleavages or the PTMs. The search results were filtered by a peptide FDR of 1%.

## SWATH sample acquisition

The SWATH data were acquired using the same mass spectrometer (TripleTOF 5600, Sciex) and LC-setup as described for the DDA samples. The mass spectrometer was operated in SWATH mode covering the mass range from 400 to 1000 Th with 75 windows (8 Th width with 1 Da overlap). The accumulation time was set to 40 ms. Additionally, an MS1 scan was acquired in the mass range from 400 to 1000 Th with an accumulation time of 250 ms, resulting in a cycle time of 3.3 s. The samples were acquired in triplicates.

## SWATH data analysis

All SWATH data were directly analyzed in Spectronaut 7.0 (Biognosys) (*Bruderer et al., 2015*) without any file conversion using the previously generated spectral library. The following settings were applied in Spectronaut 7.0: peak detection – dynamic iRT, correction factor 1; dynamic score refinement and MS1 scoring – enabled; interference correction and cross run normalization (total peak area) – enabled. Peptides were grouped according to the protein grouping by MaxQuant during generation of the spectral library. Spiked-in HRM peptides were used in the analysis for retention time and m/z calibration. The m/z tolerance was in the range of 12 ppm and the median extraction window was 4 min. All results were filtered by a Q value of 0.05 (equivalent to an FDR of 5% at the peptide level). All other settings were set to default. Protein intensities were calculated by summing the peptide peak areas (sum of fragment ion peak areas as calculated by Spectronaut) from the Spectronaut output file.

## Results

The objective of our study was to establish a workflow to elucidate peptide-resolved modification dynamics in an unbiased manner without the need for heavy isotope-labeled reference proteins or peptides. Our strategy focused on profiling unmodified peptides, as any variation in the extent of modification of a peptide results in the reduction of the amount/intensity of the remaining unmodified peptide species. This consideration also applies to modifications on or close to proteolytic cleavage sites that interfere with the proteolytic cleavage. In such cases, the affected proteolytic cleavage yield shows a reduction which is equivalent to the modification extent. This interference with the proteolytic cleavage results in a proteolytic missed cleavage peptide, which in turn leads to a change in the amount/intensity of the two unmodified, fully cleaved peptide species. Thus, variation in the amount of an unmodified peptide can be used to identify and quantify peptides whose modification state changes relative to a reference point in a time series or in a dose-dependent manner. The idea of focusing on the quantity of the unmodified peptide to identify and quantify modified peptides was implemented in the original FLEXIQuant approach, which uses stable isotope-labeled reference samples (*Singh et al., 2009*; *Singh et al., 2012a*). Here, we introduce a new, label-free (LF) implementation of this approach: FLEXIQuant-LF.

FLEXIQuant-LF enables unbiased indirect identification of differentially abundant peptides resulting from PTM in label-free mass-spectrometry data. Besides identifying differentially modified peptides the tool can also quantify the extent of these modifications. Our tool only requires raw peptide intensities as input data making it widely applicable. FLEXIQuant-LF determines differentially abundant unmodified peptides using RANSAC-based robust linear regression in combination with a

sophisticated normalization and score calculation procedure (*Figure 1B*). The RANSAC algorithm iteratively determines outliers in the data and fits the regression line only to observations not classified as outliers (inliers). This ensures that the regression line is fitted to the most robustly quantified peptides and facilitates the quantification of the modification extent based on the distance to the regression line.

We benchmarked FLEXIQuant-LF by analyzing the anaphase promoting complex/cyclosome (APC/C) during mitotic arrest. Application of FLEXIQuant-LF to this complex for proof of concept had two advantages: (1) excellent enrichment protocols for isolation of the complex by targeting the cell division cycle protein 27 (CDC27) are well established (*Singh et al., 2009*) and (2) the cell cycle-dependent phosphorylation dynamics of APC/C have been qualitatively and quantitatively described (*Singh et al., 2009*; *Steen et al., 2008*). We studied the APC/C after treating thymidine-synchronized HeLa cells with nocodazole for 4 hr, 8 hr, and 10 hr to induce prometaphase arrest. The APC/C isolated from S-phase cells served as a control (workflow in *Figure 1A*).

The software is open source and available in two versions: an easy-to-use graphical user interface (GUI) version as well as a command line interface (CLI) version to allow for integration into bioinformatics analysis pipelines.

## LC/MS analysis of the APC/C

Prior to analysis of the peptide-resolved modification dynamics of the APC/C, the complex was purified from each time point separately by co-immunoprecipitation (co-IP) using an anti-CDC27 antibody (workflow in *Figure 1A*). The protein digests of the co-immunoprecipitates from the different cell cycle stages were analyzed in triplicate using an LC-SWATH method. The SWATH dataset was analyzed using a project-specific spectral library comprising 13,724 peptides representing 2041 proteins (details in *Material and Methods*). Two out of 12 technical replicates were excluded from the analysis due to low number of identified/quantified proteins (one each from the 4 hr and 10 hr time points). As the APC/C is a well-characterized protein complex, we focused on the bona fide members of this protein complex, namely APC1, APC4, APC5, APC7, APC10, APC15, APC 16, CDC16, CDC20, CDC23, CDC26, and CDC27 (raw intensities of all peptides can be found in *Supplementary file 1*). For further FLEXIQuant-LF analysis, only unmodified peptides, and peptides with methionine oxidation or carbamidomethylation on cysteine were considered. Furthermore, any peptide had to be reliably quantified across all runs. In total, 248 out of 289 APC/C component-derived peptides (86%) met those criteria. Across these 248 quantified peptides, the median Pearson correlation coefficient between replicates was 0.98 (range: 0.96 to 0.99) highlighting excellent quantification.

## Benchmarking FLEXIQuant-LF with CDC27 and APC5

For the benchmarking of FLEXIQuant-LF, we focused on CDC27 and APC5. Cell cycle-dependent modification dynamics have been extensively studied for both proteins (*Singh et al., 2009*; *Steen et al., 2008*). These pre-existing published data enabled us to validate our peptide classifications.

We quantified 33 peptides corresponding to CDC27 (sequence coverage: 55%), of which nine peptides were classified as likely differentially modified within the time series (*Figure 2A*, magenta bars) by our FLEXIQuant-LF method. Four peptides were classified as possibly differentially modified (*Figure 2A*, blue bars). One peptide ($_{31}$LYAEVHSEEALFLLATCamYYR$_{50}$) was classified as outlier based on its raw scores and removed before RM score calculation.

We first checked for modified peptide species within our DDA dataset. In total, we identified six CDC27-derived phosphorylated peptides in the DDA data (*Figure 2A*, orange stars). The FLEXIQuant-LF analysis classified the unmodified versions of these six phosphopeptides as follows: five likely differentially modified peptides and one possibly differentially modified peptide. For the other four peptides identified as likely differentially modified as well as for two out of the three remaining peptides classified as possibly differentially modified by FLEXIQuant-LF, phosphorylations, and/or ubiquitinations were described (*Steen et al., 2008*) or deposited online at http://www.phosphosite.org (*Hornbeck et al., 2015*) (yellow stars). In summary, FLEXIQuant-LF demonstrated an excellent performance in detecting potentially modified peptides. We found evidence of modification for all

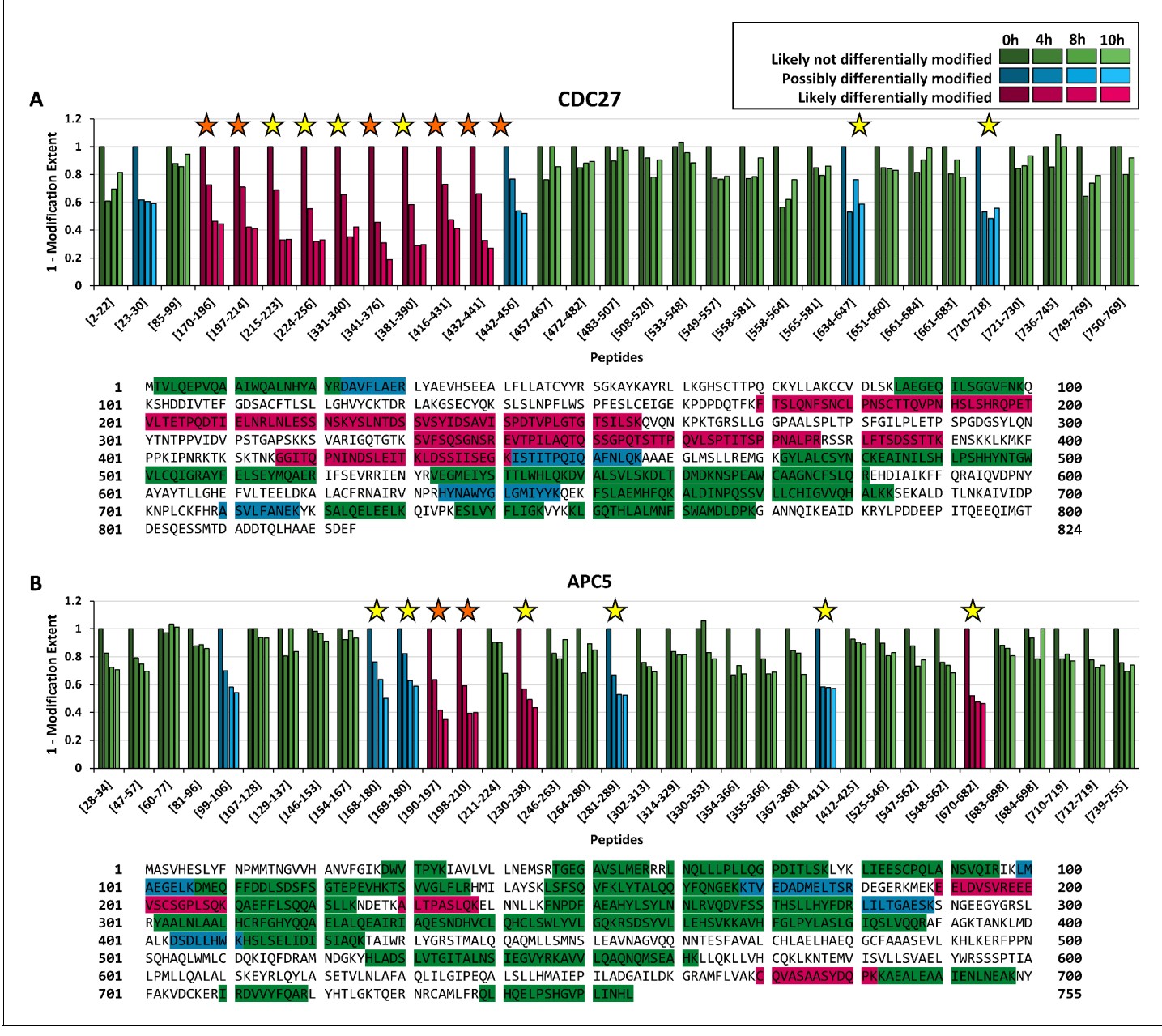

**Figure 2.** Benchmarking FLEXIQuant-LF with CDC27 and APC5. Peptide classification and extent of modification quantification of CDC27 peptides (**A**) and APC5 peptides (**B**) with protein sequences and measured peptides highlighted underneath. Green bars indicate peptides that were classified as likely *not* differentially modified (RM score ≥0.6), blue bars indicate peptides classified as possibly differentially modified (0.5 ≤ RM score<0.6) and magenta bars indicate peptides that were classified as likely differentially modified (RM score ≤0.5). Shading indicates timepoints from 0 hr (S phase; darkest shade) to 10 hr (brightest shade). Positions of peptides within the protein are given from the N- to C-terminus. Orange stars indicate modifications identified in the DDA dataset while yellow stars indicate modifications described in the literature. (**A**) FLEXIQuant-LF analysis of 32 CDC27 peptides after filtering classified nine and four peptides as likely and possibly differentially modified over the course of the experiment. We found PTM evidence for all peptides classified as likely differentially modified (for five in our DDA data and for all peptides online [see also *Table 1*]) as well as for three out of four peptides classified as possibly differentially modified (for one in our DDA data and all except of peptide [23-30] described online). (**B**) Out of 35 quantified peptides of APC5, four and five peptides were classified as likely and possibly differentially modified. We found evidence for the likely differentially modified peptides in our DDA data or as previously reported. Additionally, we found evidence for four out of five possibly differentially modified peptides online. Only for [99-106], we could not find any evidence for modification.

**Table 1.** Overview of FLEXIQuant-LF analysis for the APC/C complex components during nocodazole arrest.
Column name explanations: Quant. peptides: quantified peptides; Seq. cov.: sequence coverage; Likely diff. modified: number of peptides classified as likely differentially modified; Possibly diff. modified: number of peptides classified as possibly differentially modified; Excluded: number of peptides classified as outliers and thus excluded from RM score calculation; (DDA: number of peptides classified as differentially modified for which we found evidence in our DDA data; Lit.: number of peptides classified as differentially modified for which we found evidence for modification in the literature).

| Protein | Quant. peptides | Seq. cov. | FLEXIQuant-LF analysis | | | Evidence | | |
| | | | Likely diff. modified | Possibly diff. modified | Excluded | DDA | Lit. | No evidence |
|---|---|---|---|---|---|---|---|---|
| APC1 | 54 | 39% | 7 | 4 | 4 | 8 | 9 | 2 |
| APC4 | 28 | 43% | 0 | 0 | 2 | - | - | 0 |
| APC5 | 37 | 59% | 4 | 5 | 2 | 2 | 8 | 1 |
| APC7 | 27 | 52% | 4 | 2 | 1 | 1 | 4 | 2 |
| APC10 | 9 | 69% | 0 | 0 | 3 | - | - | 0 |
| APC15 | 1 | 30% | 0 | 0 | 0 | - | - | 0 |
| APC16 | 3 | 30% | 0 | 0 | 0 | - | - | 0 |
| CDC16 | 23 | 41% | 3 | 2 | 3 | 1 | 3 | 2 |
| CDC20 | 7 | 22% | 1 | 0 | 1 | 0 | 1 | 0 |
| CDC23 | 33 | 52% | 3 | 3 | 4 | 3 | 5 | 1 |
| CDC27 | 33 | 55% | 9 | 4 | 1 | 6 | 12 | 1 |
| *in total* | 248 | AVG 47% | 30 12% | 20 8% | 20 8% | 21 42% | 41 82% | 9 18% |

peptides identified as likely modified in our study and for three out of four possibly differentially modified peptides.

In addition to the detection of potentially modified peptides, we designed FLEXIQuant-LF to also quantify the degree of differential modification. Relative to time point 0 hr (S-phase) in our time series, the N-terminal region of CDC27 (amino acids 170–441) was increasingly modified after 4 hr (27% to 54%, average: 36%) and reached its maximum modification extent after 10 hr (between 55% and 81%, average: 65%) (*Figure 2A*). The peptide spanning amino acids 442 to 456 ($_{442}$ISTITPQIQAFNLQK$_{456}$) also showed an increasing degree of modification but to a lower extent (up to 48% modified after 10 hr).

For APC5, two peptides ($_{497}$FPPNSQHAQLWMLCamDQK$_{513}$ and $_{35}$IAVLVLLNEMSR$_{46}$) out of 37 quantified peptides (sequence coverage: 59%) were classified as outliers based on their raw scores and were excluded from further analysis. Out of the 35 remaining peptides, four were classified as likely differentially modified across the time points (*Figure 2B*, magenta bars), while five peptides were considered as possibly differentially modified (*Figure 2B*, blue bars). We identified in our DDA data the phosphorylated cognates for two of the four likely modified peptides (*Figure 2B*, orange stars). We found evidence of modifications within the literature for the other two peptides identified as likely differentially modified in our FLEXIQuant-LF analysis as well as for four of the five peptides classified as possibly differentially modified (yellow stars) (*Hornbeck et al., 2004*). Interestingly, we did not find evidence for modification for one possibly differentially modified peptide ($_{99}$LMAE-GELK$_{106}$). This could be interpreted either as a false positive classification or that this peptide has a hitherto undescribed modification on the methionine, glutamic acid or lysine residues.

Overall, these findings are in excellent agreement with the original label-based FLEXIQuant analysis of CDC27 and APC5 (*Singh et al., 2009*), which identified the same protein regions as being modified to a comparable extent. It was noted that compared to the original FLEXIQuant study the extent of modification was lower in our novel FLEXIQuant-LF analysis. A highly probable explanation for this phenomenon could be that there was a basal modification level of up to 20% for some peptides in S-phase, as shown with FLEXIQuant (*Singh et al., 2009*) which determines the absolute extent of modification. Thus, we emphasize that the FLEXIQuant-LF approach provides the extent of modification relative to the given reference time point.

## The peptide-resolved modification dynamics of other APC/C complex components

FLEXIQuant-LF has an advantage over the conventional FLEXIQuant implementation in that the analysis can easily be performed on all proteins within a sample (provided a minimum of five peptides was quantified to allow for linear regression). Here, we extended our analysis to all other APC/C proteins of which sufficient peptides were reliably quantified in our data, namely APC1 (*Figure 3*) and APC4, APC7, APC10, CDC16, CDC20, and CDC23 (*Figure 3—figure supplement 1*). By applying FLEXIQuant-LF, we classified between zero (APC4, APC10) and seven peptides (APC1) per protein as likely differentially modified and between zero (APC4, APC10, CDC20) and four peptides (APC1) per protein as possibly differentially modified (summary in *Table 1* and *Table 2*).

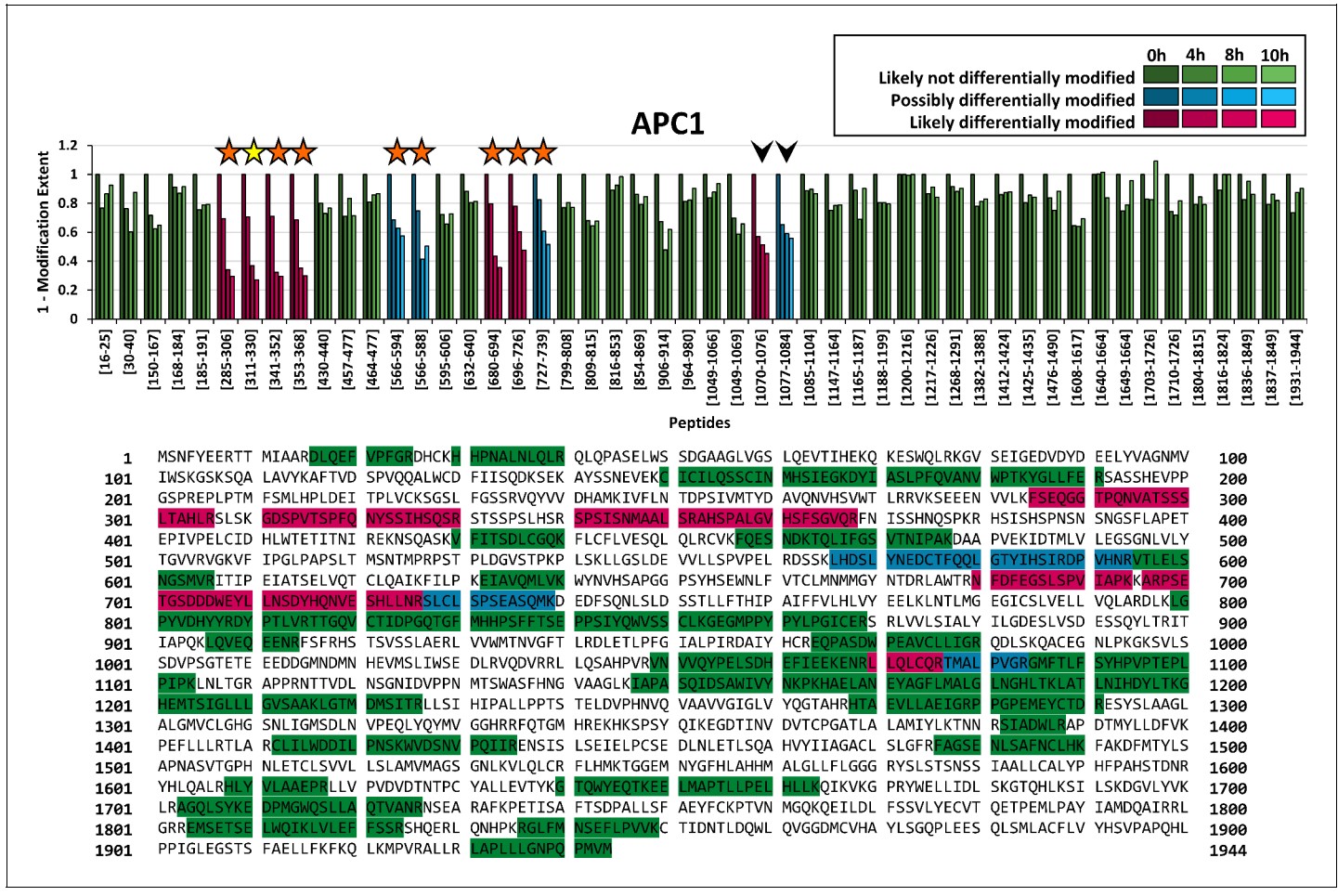

**Figure 3.** Application of FLEXIQuant-LF to remaining APC/C core components. Peptide classification and extent of modification of APC1. Seven out of 50 quantified peptides were classified as likely differentially modified and four peptides as possibly differentially modified. We found evidence of modification for six out of seven peptides classified as likely differentially modified (five in our DDA data and six described in the literature) as well as for three out of four peptides classified as possibly differentially modified in our DDA data (see also *Table 1*). Additionally, two peptides were classified as likely differentially modified and possibly differentially modified (indicated by arrow heads) but no evidence of modification was found in literature. Interestingly, these two peptides are consecutive, and the second peptide starts with threonine (AA 1077). An as of now undiscovered threonine modification such as a phosphorylation would likely lead to a highly increased missed cleavage rate and could explain the reduction of the signal intensities of both peptides. Classification and extent of modification of the remaining APC/C core components is shown in *Figure 3—figure supplements 1* and *2*.

The online version of this article includes the following figure supplement(s) for figure 3:

**Figure supplement 1.** FLEXIQuant-LF analysis of remaining APC/C components analyzed as superprotein.

**Figure supplement 2.** FLEXIQuant-LF analysis of APC/C components as superprotein.

**Table 2.** Differentially modified peptides.

Overview of peptides classified as likely or possibly differentially modified and resulting RM scores of the FLEXIQuant-LF analysis of APC/C.

| Protein | Peptide | Start | End | RM score 4 hr | RM score 8 hr | RM score 10 hr |
|---------|---------|-------|-----|---------------|---------------|----------------|
| APC1 | GDSPVTSPFQNYSSIHSQSR | 311 | 330 | 0.71 | 0.37 | 0.27 |
| | SPSISNMAALSR | 341 | 352 | 0.71 | 0.32 | 0.29 |
| | FSEQGGTPQNVATSSSLTAHLR | 285 | 306 | 0.69 | 0.34 | 0.30 |
| | AHSPALGVHSFSGVQR | 353 | 368 | 0.69 | 0.35 | 0.30 |
| | NFDFEGSLSPVIAPK | 680 | 694 | 0.80 | 0.43 | 0.36 |
| | LLQLCamQR | 1070 | 1076 | 0.57 | 0.51 | 0.45 |
| | ARPSETGSDDDWEYLLNSDYHQNVESHLLNR | 696 | 726 | 0.78 | 0.61 | 0.48 |
| | LHDSLYNEDCamTFQQLGTYIHSIR | 566 | 588 | 0.75 | 0.42 | 0.50 |
| | SLCamLSPSEASQMK | 727 | 739 | 0.83 | 0.61 | 0.51 |
| | TMALPVGR | 1077 | 1084 | 0.65 | 0.59 | 0.56 |
| | LHDSLYNEDCamTFQQLGTYIHSIRDPVHNR | 566 | 594 | 0.69 | 0.63 | 0.57 |
| APC5 | EELDVSVR | 190 | 197 | 0.64 | 0.42 | 0.35 |
| | EEEVSCamSGPLSQK | 198 | 210 | 0.59 | 0.39 | 0.40 |
| | ALTPASLQK | 230 | 238 | 0.57 | 0.49 | 0.43 |
| | CamQVASAASYDQPK | 670 | 682 | 0.52 | 0.47 | 0.46 |
| | KTVEDADMELTSR | 168 | 180 | 0.76 | 0.64 | 0.50 |
| | LILTGAESK | 281 | 289 | 0.67 | 0.53 | 0.52 |
| | LMAEGELK | 99 | 106 | 0.70 | 0.58 | 0.54 |
| | DSDLLHWK | 404 | 411 | 0.58 | 0.58 | 0.57 |
| | TVEDADMELTSR | 169 | 180 | 0.82 | 0.63 | 0.59 |
| APC7 | VRPSTGNSASTPQSQCamLPSEIEVK | 116 | 139 | 0.85 | 0.46 | 0.41 |
| | AYAFVHTGDNSR | 242 | 253 | 0.60 | 0.47 | 0.47 |
| | DMAAAGLHSNVR | 43 | 54 | 0.61 | 0.52 | 0.49 |
| | YTMALQQK | 100 | 107 | 0.70 | 0.47 | 0.54 |
| | ALTQRPDYIK | 468 | 477 | 0.67 | 0.55 | 0.57 |
| CDC16 | QTAEETGLTPLETSR | 573 | 587 | 0.86 | 0.34 | 0.34 |
| | CamYDFDVHTMK | 544 | 553 | 0.60 | 0.55 | 0.45 |
| | SSICamLLR | 130 | 136 | 0.65 | 0.60 | 0.49 |
| | IYDALDNR | 139 | 146 | 0.71 | 0.56 | 0.52 |
| | DPFHASCamLPVHIGTLVELNK | 261 | 280 | 0.44 | 1.00 | 0.60 |
| CDC20 | VLSLTMSPDGATVASAAADETLR | 446 | 468 | 0.51 | 0.32 | 0.34 |
| CDC23 | NQGETPTTEVPAPFFLPASLSANNTPTR | 558 | 585 | 0.82 | 0.27 | 0.13 |
| | RVSPLNLSSVTP | 586 | 597 | 0.71 | 0.30 | 0.20 |
| | VSPLNLSSVTP | 587 | 597 | 0.82 | 0.36 | 0.28 |
| | AALYFQR | 350 | 356 | 0.67 | 0.57 | 0.52 |
| | LWDEASTCamAQK | 525 | 535 | 0.74 | 0.62 | 0.56 |
| | NTSAAIQAYR | 380 | 389 | 0.70 | 0.55 | 0.56 |
| CDC27 | EVTPILAQTQSSGPQTSTTPQVLSPTITSPPNALPR | 341 | 376 | 0.46 | 0.31 | 0.19 |
| | LDSSIISEGK | 432 | 441 | 0.66 | 0.33 | 0.27 |
| | LFTSDSSTTK | 381 | 390 | 0.58 | 0.29 | 0.30 |
| | YSLNTDSSVSYIDSAVISPDTVPLGTGTSILSK | 224 | 256 | 0.56 | 0.32 | 0.33 |

*Table 2 continued on next page*

*Table 2 continued*

| Protein | Peptide | Start | End | RM score | | |
|---------|---------|-------|-----|------|------|-------|
| | | | | 4 hr | 8 hr | 10 hr |
| | LNLESSNSK | 215 | 223 | 0.69 | 0.33 | 0.33 |
| | GGITQPNINDSLEITK | 416 | 431 | 0.73 | 0.47 | 0.41 |
| | QPETVLTETPQDTIELNR | 197 | 214 | 0.71 | 0.42 | 0.41 |
| | SVFSQSGNSR | 331 | 340 | 0.66 | 0.35 | 0.42 |
| | FTSLQNFSNCamLPNSCamTTQVPNHSLSHR | 170 | 196 | 0.73 | 0.46 | 0.45 |
| | ISTITPQIQAFNLQK | 442 | 456 | 0.77 | 0.54 | 0.52 |
| | ASVLFANEK | 710 | 718 | 0.53 | 0.48 | 0.56 |
| | HYNAWYGLGMIYYK | 634 | 647 | 0.53 | 0.76 | 0.59 |
| | DAVFLAER | 23 | 30 | 0.62 | 0.61 | 0.59 |

From APC1, we quantified 54 peptides (sequence coverage: 39%) in total, of which four peptides were classified as outlier based on their raw scores and were excluded from further analysis ($_{930}$LVVWMTNVGFTLR$_{942}$, $_{943}$DLETLPFGIALPIR$_{956}$, $_{1546}$TGGEMNYGFHLAHHMALGLLFLGGGR$_{1571}$ and $_{1618}$LLVPVDVDTNTPCamYALLEVTYK$_{1639}$). Out of the remaining 50 peptides, seven peptides were classified as likely differentially modified and four peptides as possibly differentially modified (*Figure 3*, magenta and blue bars, respectively). We identified phosphorylated peptides in the DDA data for five of the seven likely and for three out of four possibly differentially modified peptides (*Figure 3*, orange stars).

We also classified the peptide $_{311}$GDSPVTSPFQNYSSIHQSR$_{330}$ as likely differentially modified by our FLEXIQuant-LF analysis, although no modified cognate peptides were identified in the DDA data. However, multiple modifications, that is phosphorylations on S313, T316, S317, Y322 and S324, have been reported in previous studies (*Hornbeck et al., 2015*) (indicated by a yellow star in *Figure 3*). For one likely ($_{1070}$LLQLCamQR$_{1076}$) and one possibly differentially modified peptide ($_{1077}$TMALPVGR$_{1084}$), we could not find published evidence of PTMs within the sequence of those two peptides. Interestingly, these are consecutive peptides with the latter one starting with a threonine residue (AA 1077). A hitherto undiscovered phosphorylation at this threonine residue could lead to a highly increased missed cleavage which would be consistent with the observed intensity reduction of both peptides.

A more detailed analysis of the peptide-resolved modification dynamics observed in APC1 identified two regions within the protein that appear to have different modification kinetics (*Figure 3*). Peptides in the N-terminal part ranging from residues 285–368 (four peptides), showed a degree of modification of 29–31% after 4 hr, 63–68% after 8 hr and between 70% and 71% after 10 hr. In contrast, the peptides in a more C-terminal domain spanning residues 680–739 (three peptides) showed slower modification kinetics and a lower degree of modification at 10 hr. After 4 hr, 17–22% change in modification status was observed. The extent of modification changed between 39% and 57% after 8 hr, and 49–64% after 10 hr. These data demonstrate that tools to study the peptide-resolved modification dynamics in a quantitative manner will greatly improve the current understanding of biological processes.

## Testing and validating reproducibility of FLEXIQuant-LF

We tested the reproducibility of FLEXIQuant-LF by analyzing a large inhouse MRM data set consisting of 81 samples of which 26 were reference group samples. Over 1000 runs FLEXIQuant-LF yielded ambiguous results for 6 out of 81 samples when using one RANSAC initiation. In all six cases, two distinct outcomes were observed. To improve the reproducibility, we initiated the RANSAC algorithm multiple times within a single run (1, 10, 20, 30, 40, 50, 60, 70, 80, 90, and 100) and selected the best model (based on r2 score). This was again tested by running each set of initiations 1000 times and analyzing the slopes of the resulting best models. Initiating RANSAC 10 times resulted in six ambiguous samples, 20 initiations in 5, 30 in 2 and 40 in 3. Between 50 and 100 initiations, the number of ambiguous samples was oscillating between 1 and 2 (*Figure 4A*). Furthermore, as

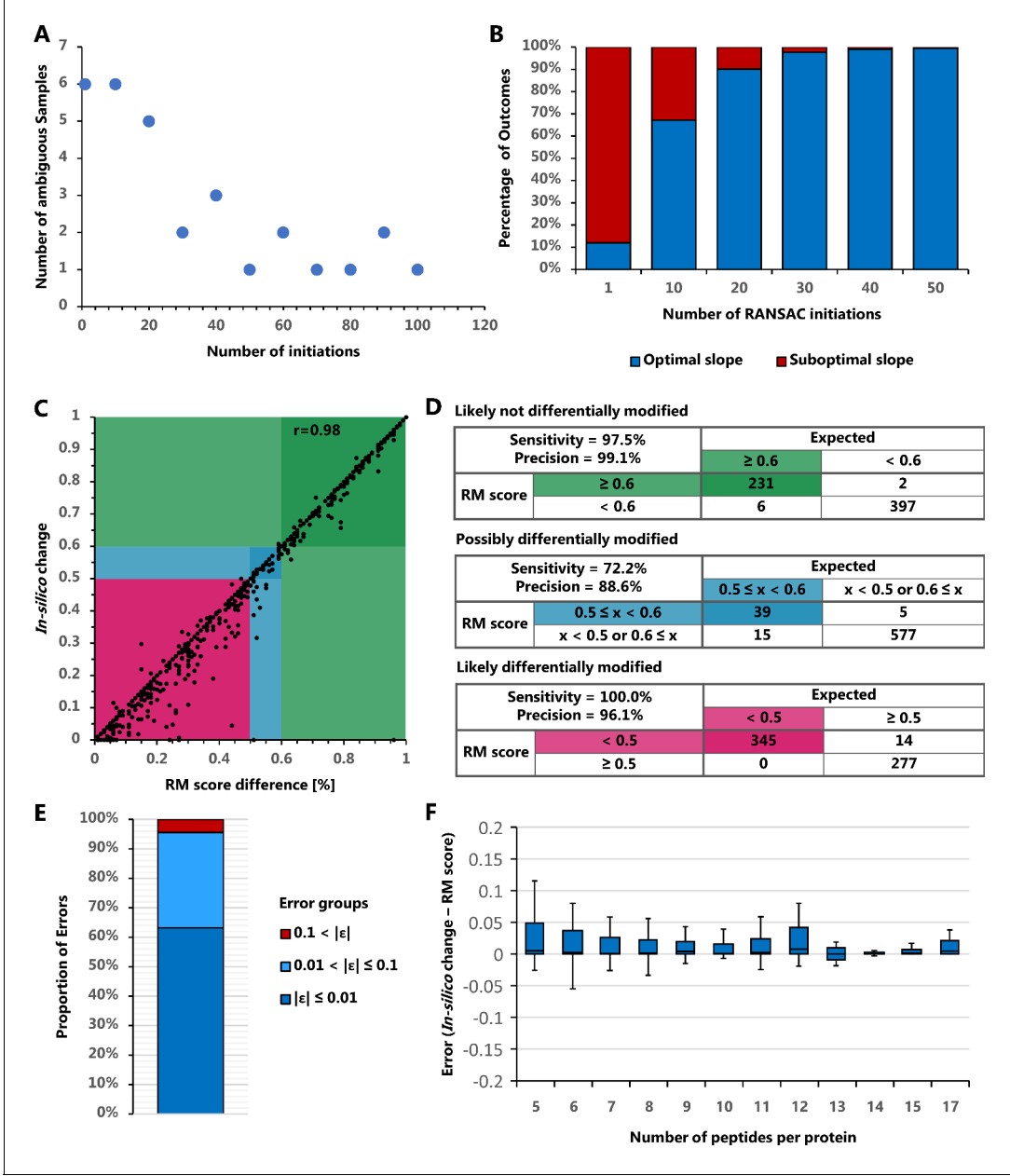

**Figure 4.** FLEXIQuant-LF reproducibility and error estimation. (**A**) Number of ambiguous samples with different number of RANSAC initiations. The number of samples with ambiguous results over 1000 runs decreases with increasing number of RANSAC initiations but starts to oscillate between one and two ambiguous samples from 50 initiations upwards. (**B**) Fraction of optimal and suboptimal outcomes of PMF with sample 30. The frequency of a suboptimal outcome decreases with increasing number of RANSAC initiations. (**C**) Correlation of the expected/in silico created change with the RM score difference before and after the in silico change. Expected and measured changes highly correlate (Pearson r = 0.98). (**D**) Classification error estimation using definitions of likely not, possibly, and likely differentially modified as described in *Figure 1B*. With a sensitivity and precision of 72.2% and 88.6%, respectively, the transition area of possibly differentially modified evaluates expectedly worse than the likely not and likely differentially modified classifications (both with sensitivity and precision >96%). (**E**) Cumulative quantification error frequencies show 94.7% of cases below 0.1. (**F**) Quantification error associated with number of peptides used per protein (boxes indicated 2nd and 3rd quartiles while whiskers indicate data 1.5 times the inter quartile range above and below these quartiles). Overall quantification errors are very low and further improve with the number of peptides used for FLEXIQuant-LF analysis.

illustrated by means of sample 30, the frequency of different outcomes within a sample decreased with increasing number of RANSAC initiations (*Figure 4B*). With one initiation the suboptimal outcome prevailed (88%), whereas with 10 initiations the optimal outcome was predominant (67.2%). Using 30 initiations the fraction of suboptimal outcomes was 2.2% and further decreased to 0.5% with 50 initiations.

Overall, using 30 RANSAC initiations and selecting the best model resulted in 44 suboptimal outcomes in 81,000 runs of FLEXIQuant-LF (0.05%). These results confirm the benefits of multiple RANSAC initiations and demonstrate a very high reproducibility of the developed approach. Based on these finding, the default number of RANSAC initiations in FLEXIQuant-LF was set to 30 but can be changed by the user.

We also tested the quality of FLEXIQuant-LF classification and quantification on an independent DIA dataset (*Bruderer et al., 2017*). After stringent filtering 122 proteins without detected modifications and with unique and unambiguous associations to peptides were selected for in silico modification, that is intensities of single peptides were reduced to simulate the presence of modified peptide species. For each protein, four peptides were randomly chosen, and their intensity adjusted by a random change factor for each replicate resulting in 1574 total FLEXIQuant-LF runs (*Supplementary file 3*). RM score changes showed a high correlation with the in-silico intensity change factors (Pearson's r = 0.98, *Figure 4C*). The overall near perfect correlation of the in-silico and RM score changes was also reflected in the classification results. The classes of likely not differentially modified and likely differentially modified peptides also showed near optimal performance with sensitivity and precision values at 97.5% and 99.1%, and 100.0% and 96.1%, respectively, while the class of possibly differentially modified peptides with hard-to-define borders showed reduced performance with a sensitivity of 77.2% and precision of 88.6% (*Figure 4D*). This decrease in accuracy likely reflects more on the definition criteria of the middle group than on the performance of the method. Additionally, of interest was the robustness of FLEXIQuant-LF results in relation to the number of peptides available for each protein. Overall, quantification errors were low with a mean error of 0.020 (SD = 0.058,<0.1 for 94.7% of cases, *Figure 4E*). Quantification errors in proteins with fewer peptides (n = 5–7: μ = 0.027, SD = 0.072) did not deviate from proteins with more peptides (n = 8–10: μ = 0.010, SD = 0.020; n = 11–13: μ = 0.013, SD = 0.026; n = 14–17: μ = 0.006, SD = 0.012) and were within a very small error margin of modification extent of <5% (*Figure 4F*). Outliers are rare (<5%) and can be attributed to the random selection and simulation effects of the assessment approach.

## Analyzing multiple proteins as a 'superprotein'

For the two smallest APC/C subunits, namely APC15 and APC16, the number of reliably quantified peptides were 1 and 3, respectively, that is below the described threshold of five quantifiable peptides, making the above described FLEXIQuant-LF analysis nonapplicable. However, if proteins are known to have invariable expression/abundance profiles over the considered time period or between different conditions of interest, as for example observed for the core components of the APC/C, we hypothesized that the peptides of multiple proteins can be analyzed together by treating this set of proteins with invariable abundance as one 'superprotein'. This would expand the FLEXIQuant-LF approach to smaller proteins and allow for the identification and quantification of differential modification of proteins with less than five quantified peptides that could not be analyzed by the normal FLEXIQuant-LF implementation otherwise.

To test this 'superprotein' approach, we applied FLEXIQuant-LF to analyze all reliably quantified APC/C proteins in our data in a combined fashion as a 'superprotein', namely APC1, APC4, APC5, APC7, APC10, APC15, APC16, CDC16, CDC23, and CDC27. CDC20 was excluded from this 'superprotein' approach as it is known that this APC/C regulator displays changing affinity to the APC/C during the M-phase of the cell cycle, that is its abundance within the APC/C changes during the investigated time period (*Fang et al., 1999*). We validated the 'superprotein' strategy by comparing its results to the results of the FLEXIQuant-LF analysis of the individual proteins for all proteins that could be analyzed with both strategies. The two approaches resulted in identical classification of peptides as well as near identical quantification of the modification extent (mean RM score difference = 2.45E-10, $r^2$ = 1), demonstrating the applicability of the 'superprotein' strategy (*Supplementary file 2*).

The FLEXIQuant-LF analysis of APC15 and APC16 using the 'superprotein' approach revealed that all four peptides associated with these two small proteins not amenable to the conventional FLEXIQuant-LF approach are likely not differentially modified (*Supplementary file 2* and *Figure 3— figure supplement 2A and B*). These results are in accordance with the literature where, to the best of our knowledge, no cell cycle-dependent differential modifications in APC15 and APC16 have been described for the detected protein domains (*Hornbeck et al., 2015*).

## Discussion

Our FLEXIQuant-LF is based on robust linear regression, as implemented in the RANSAC algorithm. In general, linear regression requires a certain number of data points to yield meaningful results and becomes more accurate with increasing number of data points. Thus, FLEXIQuant-LF becomes more powerful with increasing numbers of unique peptides that can be identified for the proteins of interest. We set the minimum number of unique peptides required for a FLEXIQuant-LF analysis to five, although a larger number of peptides will provide more robust results. Addressing this limitation, we introduce the 'superprotein' approach.

In the presented proof-of-concept study, we applied FLEXIQuant-LF to a time series experiment. However, FLEXIQuant-LF can also be applied to study protein modification differences between different experimental conditions or cohorts in general, e.g. comparing diseased vs. healthy individuals or testing the effect of different drugs. As FLEXIQuant-LF only determines the extent of differential modification of a peptide relative to the reference based on the unmodified counterparts, quantification of the absolute abundance of the modification still requires an isotopically labeled internal standard. However, using a reference time point or condition has the advantage of highlighting modifications that exhibit dynamic regulation in the context of interest. Furthermore, FLEXIQuant-LF can be used to repeatedly analyze the same data set with varying reference samples, providing complementary perspectives on the peptide-resolved modification dynamics. This consideration is particularly important for the current implementation of FLEXIQuant-LF which only quantifies modifications extents that are increasing relative to the reference sample. This is due to our normalization method used for RM score calculation, which excludes peptides with decreased modification. FLEXIQuant-LF reports these peptides in the '_removed_peptides.csv' output file and raw score information can be retrieved from the '_raw_score.csv' output file. We reasoned that the reference sample is normally the least modified protein. In case this assumption does not hold, a second FLEXIQuant-LF can be carried out after selecting another sample as the reference.

While FLEXIQuant-LF, similar to the initial FLEXIQuant approach, indirectly identifies peptides that are differentially modified, identifying the specific type and site of modification is not the objective of this analysis. The detailed characterization and localization of the modifications require either more in-depth data analysis or follow-up experiments. Nevertheless, this analysis method can be easily applied in any affinity purification or global proteomics experiment and the results can further be used to identify proteins to be synthesized as reference standards, if more in-depth experiments are needed. In the original FLEXIQuant approach, changes in the modification extent had to be above 20% in order to be reliably detected. For our proof of concept study, we chose 40% which is consistent with stoichiometries commonly observed in cell cycle-focused or signaling pathway activation-related large-scale phosphoproteomics studies, which reported serine and threonine phosphorylation extents exceeding 75% and tyrosine phosphorylation extents above 50% (*Sharma et al., 2014*). However, our FLEXIQuant-LF approach enables the selection of a minimum change in modification extent deemed to be interesting and/or compatible with the precision of the dataset; clearly the usual trade-off considerations between specificity and sensitivity apply.

Even though APC/C is a moderately abundant complex within the cell, we opted for affinity enrichment to ensure a high number of unique peptides per protein of the complex components are measured. For highly abundant protein classes such as the ribosome and proteasome (*Beck et al., 2011*), a direct FLEXIQuant-LF analysis on the whole cell lysate is most likely a viable option. However, we expect that future technological developments in LC/MS instrumentation and data analysis approaches will make less abundant proteins and protein complexes such as signaling proteins/complexes amenable to FLEXIQuant-LF analysis without further enrichment of the proteins/protein complexes of interest.

A premise of the FLEXIQuant-LF method is, that the modification state of the majority of peptides in a protein does not change between experimental conditions or over the time period considered in a study. This is likely to hold true for most proteins. However, in some cases with more modified than unmodified peptides as in Tau for example (*Mair et al., 2016*), FLEXIQuant-LF might not be applicable.

The combination of affinity purification of the protein complex and DIA analysis facilitated the precise and reproducible peptide quantification. Currently, DIA is the most suitable acquisition method due to its precise peptide quantification capabilities with fewer missing values. However, the FLEXIQuant-LF approach is also applicable to other label-free quantification methods.

## Conclusions

Here, we presented FLEXIQuant-LF, a computational label-free implementation of the original FLEXI-Quant method using a DIA dataset of the APC/C during mitotic arrest as a proof of concept. FLEXI-Quant-LF is a widely applicable analysis workflow building on the previously published FLEXIQuant idea of monitoring the unmodified peptides and changes to their signal intensities to indirectly detect modification events and to quantify the degree of modifications. The use of a reference time-point or reference sample within a study instead of labeled reference proteins allows FLEXIQuant-LF to be universally applicable post hoc to any protein robustly identified across the sample series without the need for the addition of heavy isotope-labeled protein standard(s).

The application of FLEXIQuant-LF to a DIA dataset of the APC/C isolated from HeLa cells at various time points during a mitotic arrest, allowed us to recapitulate the peptide-resolved modification dynamics previously observed with our original heavy isotope-labeled internal standard-based FLEXI-Quant method. Using FLEXIQuant-LF, we easily extended our analysis to other APC/C subunits present in the sample set and delivered a comprehensive report on the peptide-resolved modification dynamics of the core components of APC/C following nocodazole-induced mitotic arrest. Furthermore, we introduced a 'superprotein' approach which allowed us to include proteins, which provided insufficient numbers of unique peptides to be analyzed individually; this is a limitation that applies in particular to small proteins, but can also be encountered in very acidic or hydrophobic proteins with few proteolytic cleavage sites. This 'superprotein' strategy enabled us to extract more information from data and gain additional insights about peptides/proteins that would have been disregarded otherwise.

FLEXIQuant-LF is unbiased toward the type of modification and relies on identification of the unmodified counterpart of a modified peptide to detect and quantify modification events, irrespective of the modification's nature, that is the types of modifications involved remain unknown. The valuable information of modification event detection and modification extent quantification can subsequently be used to study rare PTMs and to potentially discover novel PTMs using more targeted approaches in future experiments. Therefore, FLEXIQuant-LF can pave the way to a better general understanding of peptide-resolved PTM dynamics of proteins. We expect that FLEXIQuant-LF will enable researchers to revisit their data and identify novel modification-driven features of health and disease that could be used as targets for treatment.

## Data and software availability

Data are available via ProteomeXchange with identifier PXD018411.

FLEXIQuant-LF is available with graphical user interface (GUI) as standalone executable EXE file. Additionally, a command line interface (CLI) version is available as PY file. Both versions as well as the Python source code can be downloaded via GitHub: https://github.com/SteenOmicsLab/FLEXIQuantLF.

## Acknowledgements

We thank Benoit Fatou and Mukesh Kumar for providing test data during development and testing of FLEXIQuant-LF. Furthermore, we thank Kyle Higgins and Patrick van Zalm for testing the FLEXI-Quant-LF software.

## Additional information

### Funding

| Funder | Grant reference number | Author |
|---|---|---|
| National Institutes of Health | S10OD0107060 | Hanno Steen |
| National Institutes of Health | R01CA196703 | Hanno Steen |
| National Institutes of Health | R01AI099204 | Hanno Steen |
| National Institutes of Health | U01AI124284 | Hanno Steen |
| National Institutes of Health | R01NS066973 | Hanno Steen |
| National Institutes of Health | RC4GM096319 | Hanno Steen |
| National Institutes of Health | R01GM112007 | Judith A Steen |
| Deutsche Forschungsgemeinschaft | RE3474/2-2 | Bernhard Renard |

The funders had no role in study design, data collection and interpretation, or the decision to submit the work for publication.

### Author contributions

Christoph N Schlaffner, Conceptualization, Data curation, Formal analysis, Supervision, Validation, Investigation, Visualization, Methodology, Writing - review and editing; Konstantin Kahnert, Data curation, Software, Formal analysis, Validation, Investigation, Visualization, Methodology, Writing - original draft; Jan Muntel, Investigation, Visualization, Writing - original draft; Ruchi Chauhan, Investigation; Bernhard Y Renard, Conceptualization, Supervision, Funding acquisition, Validation, Methodology, Writing - review and editing; Judith A Steen, Conceptualization, Resources, Funding acquisition; Hanno Steen, Conceptualization, Resources, Supervision, Funding acquisition, Validation, Methodology, Project administration, Writing - review and editing

### Author ORCIDs

Christoph N Schlaffner  https://orcid.org/0000-0003-2717-3406
Konstantin Kahnert  https://orcid.org/0000-0002-8454-4894
Jan Muntel  https://orcid.org/0000-0003-2320-5829
Hanno Steen  https://orcid.org/0000-0003-0179-6648

### Decision letter and Author response

Decision letter https://doi.org/10.7554/eLife.58783.sa1
Author response https://doi.org/10.7554/eLife.58783.sa2

## Additional files

### Supplementary files

• Supplementary file 1. Raw peptide intensities of all identified APC/C components.

• Supplementary file 2. Comparison RM scores proteins analyzed individually and using the 'super-protein' approach.

• Supplementary file 3. Comparison RM scores for simulated independent dataset.

• Transparent reporting form

### Data availability

MS data are available via ProteomeXchange under accession code PXD018411 and PXD005573. All data generated or analysed during this study are included in the manuscript and supporting files. FLEXIQuant-LF is available with graphical user interface (GUI) as standalone executable EXE file. Additionally, a command line interface (CLI) version is available as PY file. Both versions as well as

the Python source code can be downloaded via GitHub: https://github.com/SteenOmicsLab/FLEXI-QuantLF (copy archived at https://archive.softwareheritage.org/swh:1:rev:4ea3945f86ba477227-c89e9ced75fc23751355ac/).

The following dataset was generated:

| Author(s) | Year | Dataset title | Dataset URL | Database and Identifier |
|---|---|---|---|---|
| Schlaffner CN, Kahnert K, Muntel J, Chauhan R, Renard BY, Steen JA, Steen H | 2020 | FLEXIQuant-LF: Robust Regression to quantify protein modification extent in label-free proteomics data | http://proteomecentral.proteomexchange.org/cgi/GetDataset?ID=PXD018411 | ProteomeXchange, PXD018411 |

The following previously published dataset was used:

| Author(s) | Year | Dataset title | Dataset URL | Database and Identifier |
|---|---|---|---|---|
| Bruderer R, Bernhardt OM, Gandhi T, Xuan Y, Sondermann J, Schmidt M, Gomez-Varela D, Reiter L | 2017 | Optimization of Experimental Parameters in Data-Independent Mass Spectrometry Significantly Increases Depth and Reproducibility of Results | http://proteomecentral.proteomexchange.org/cgi/GetDataset?ID=PXD005573 | ProteomeXchange, PXD005573 |

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
