## [Decision Letter]

**Acceptance summary:**

While large-scale identification of post-translational modifications by mass spectrometry has become trivial, determining the extent of these modifications remains a challenge. FLEXIQuant-LF permits an assessment of the extent of PTMs without the use of heavy labeled standards and should be especially useful in large-scale experiments involving cellular dynamics or perturbation experiments.

**Decision letter after peer review:**

Thank you for submitting your article "FLEXIQuant-LF: Robust Regression to Quantify Protein Modification Extent in Label-Free Proteomics Data" for consideration by *eLife*. Your article has been reviewed by three peer reviewers, one of whom is a member of our Board of Reviewing Editors, and the evaluation has been overseen by Philip Cole as the Senior Editor. The reviewers have opted to remain anonymous. The reviewers have discussed the reviews with one another and the Reviewing Editor has drafted this decision to help you prepare a revised submission.

The manuscript describes a method for quantifying modifications of peptides through analysis of intensity of unmodified peptides. The authors highlight that it is capable of large-scale identification and quantification of differentially modified peptides without any prior knowledge of the type of modification. The method was applied to DIA data from an experiment designed to study APC/C complex to benchmark and to identify differentially modified peptides in a time course setting. The obvious advantage of this method is that it does not rely on heavy isotope-labeled proteins/peptides. The manuscript describes a novel method, which is likely to be impactful for the field and could be a good fit as an *eLife* Tools and Resources paper. However, the way the manuscript is currently written is quite confusing in several places. The authors are advised to submit a revised manuscript that addresses the following issues:

Essential revisions:

1) Modifying the text and the Materials and methods section to emphasize that changes are measured in unmodified peptides and that the modification is not directly measured at all. Perhaps, the authors could consider including a graphic to illustrate this.

2) The authors assume that the number of molecules of each unique peptide derived from a protein will be equal. Because in routine proteomic analyses, this is not always true possibly due to the different degree of digestions at different proteolytic sites. The authors should address this issue.

3) Since many peptides often contain multiple potential sites of modifications (e.g. several phosphorylation sites on the same peptide), it is not possible to accurately measure the phosphorylation dynamics for each site. The authors should modify their claims and discuss this important caveat.

4) Although the authors state that FLEXIQuant-LF allows quantification of the modification extent without prior knowledge of the type of modification, they should explicitly state that after this analysis, the exact type of modification is still large unknown.

5) The authors should carry out an artificial positive control experiment using synthetic peptides to provide better understanding to readers of how well the algorithm works.

6) The authors should apply FLEXIQuant-LF to one of publicly available label-free phosphoproteome datasets with known answers regarding the extent of phosphorylation? If the authors can do this and obtain similar conclusions, then it can indeed be presented as a powerful strategy that can be applied in many studies.

---

## [Author Response]

Essential revisions:1) Modifying the text and the Materials and methods section to emphasize that changes are measured in unmodified peptides and that the modification is not directly measured at all. Perhaps, the authors could consider including a graphic to illustrate this.

We thank the reviewer for the comment and the suggestion to include an illustration to clarify the methodology. We have adjusted the descriptions throughout the text and the Materials and methods section to clarify that FLEXIQuant-LF, like FLEXIQuant, measures the modification extent indirectly by observing the abundance changes of the unmodified peptide counterparts. We also expanded Figure 1B to illustrate better that the unmodified peptide abundances are used to measure the modification extent.

Changes in manuscript:

**“**The development of FLEXIQuant (Full-Length-Expressed Stable Isotope-labeled Proteins for Quantification) (Singh et al., 2009; Singh et al., 2012 ), for the first time, enabled the quantification of the extent of modification across a whole protein without prior knowledge of modification identities using an indirect approach of quantifying only the unmodified peptides. FLEXIQuant relies on the principle that the total number of molecules of a given protein is conserved in a sample, thus the number of molecules of each unique peptide derived from that protein will be equal. If a peptide is modified chemically by a PTM this would reduce the abundance of its unmodified cognate in the peptide pool. […] The advantage of this approach is the precise and accurate quantification of the degree of abundance reduction due to a modification as defined by Wold (Wold, 1981) or amino acid substitutions of each quantified unmodified peptide and the ability to calculate the absolute concentration.”

Changes in manuscript:

“Here, we introduce FLEXIQuant-LF as an unbiased, label-free computational tool to indirectly detect modified peptides and to quantify the degree of modification based solely on the unmodified peptide species building upon the FLEXIQuant (Singh et al., 2009) idea developed in our lab. We developed this approach to identify and elucidate differential protein modification extent in commonly analyzed time series or case-control studies.”

Changes in manuscript:

“FLEXIQuant-LF indirectly identifies modified peptides by means of pinpointing unmodified peptides whose intensities strongly deviate from those of a reference sample using robust linear regression. […] More specifically, the FLEXIQuant-LF algorithm trains a linear regression model for each sample in the input file using the random sample consensus (RANSAC) algorithm (Fischler et al., 1981) to identify outliers iteratively and fit the model only based on the inliers, i.e. based on stably and consistently quantified unmodified peptides.”

Changes in manuscript:

“To increase reproducibility and enhance the fit of the linear regression model to the data the algorithm is executed 30 times and the best model, based on *r*^2^ scores, is selected (Figure 4A and B). […] RM scores are calculated by dividing each inlier raw score by the median of the three highest raw scores (after removing outliers) for each sample, corresponding to one minus the extent of modification as defined by FLEXIQuant (Figure 1B4).”

Changes in manuscript:

“Peptides were then classified in three categories based on their RM scores at time point 10h: (1) RM score < 0.5: peptide is likely differentially modified (magenta bars), (2) 0.5 ≤ RM score < 0.6: peptide is possibly differentially modified (blue bars) and (3) RM score ≥ 0.6: peptide is likely not differentially modified (green bars) (Figure 1B5).”

Changes in manuscript:

“The objective of our study was to establish a workflow to elucidate peptide-resolved modification dynamics in an unbiased manner without the need for heavy isotope-labeled reference proteins or peptides. […] This interference with the proteolytic cleavage results in a proteolytic missed cleavage peptide, which in turn leads to a change in the amount/intensity of the two unmodified, fully cleaved peptide species.”

Changes in manuscript:

“FLEXIQuant-LF enables unbiased indirect identification of differentially abundant peptides resulting from post-translational modification in label-free mass-spectrometry data. […] This ensures that the regression line is fitted to the most robustly quantified peptides and facilitates the quantification of the modification extent based on the distance to the regression line.”

Changes in manuscript:

“However, FLEXIQuant-LF can also be applied to study protein modification differences between different experimental conditions or cohorts in general, e.g. comparing diseased vs. healthy individuals or testing the effect of different drugs. As FLEXIQuant-LF only determines the extent of differential modification of a peptide relative to the reference based on the unmodified counterparts, quantification of the absolute abundance of the modification still requires an isotopically labeled internal standard.”

“While FLEXIQuant-LF, similar to the initial FLEXIQuant approach, indirectly identifies peptides that are differentially modified, identifying the specific type and site of modification is not the objective of this analysis.”

Changes in manuscript:

“FLEXIQuant-LF is a widely applicable analysis workflow building on the previously published FLEXIQuant idea of monitoring the unmodified peptides and changes to their signal intensities to indirectly detect modification events and to quantify the degree of modifications. The use of a reference timepoint or reference sample within a study instead of labeled reference proteins allows FLEXIQuant-LF to be universally applicable post hoc to any protein robustly identified across the sample series without the need for the addition of heavy isotope-labeled protein standard(s).”

Changes in manuscript:

“Figure 1: Workflow and FLEXIQuant-LF Concept. […] (B1) Firstly, intensities of unmodified peptides are used to indirectly identify and quantify the modification extent of a protein using the following steps. ”

2) The authors assume that the number of molecules of each unique peptide derived from a protein will be equal. Because in routine proteomic analyses, this is not always true possibly due to the different degree of digestions at different proteolytic sites. The authors should address this issue.

We thank the reviewer for the comment. We agree that different degrees of digestion at proteolytic cleavage sites could theoretically pose an issue when comparing the total amount of peptides within a protein. However, it is our experience that similar samples digested within a batch on the same day show very similar proteolysis yields. Furthermore, under non-limiting digestions conditions, the digestion yields reach a plateau, i.e. even longer digestion times will not majorly affect the ratio of fully cleaved vs. missed cleaved. Both these assumptions are also made in almost all targeted LC/MS assays which add a heavy proteotypic peptide as internal standard to the sample prior to digestion in order to absolutely quantify the protein of interest. In summary, as long as non-limiting digestion conditions are ensured, we strongly believe that the risk of vastly different degrees of digestions can be managed, especially when it is ensured that the reference sample and the samples of interest are digested in the same batch.

In addition to the biochemical aspect, FLEXIQuant-LF is sufficiently robust to add another ‘layer of protection’ against different degrees of digestion by using robust RANSAC regression line fitting: Differences in reproducibility due to e.g. variable degree of digestion does not affect the robust RANSAC linear regression fitting and will not result in biased results. This notion is underscored by the fact that we observed several pairs of fully cleaved peptides and peptides with missed cleavages – both show very similar ratios relative to the reference sample.

Of note: This treatment of proteolytic cleavages as modifications is consistent with the 1981 definition of modification by Wold (Wold, 1981) which defines it as affecting peptide bonds, carboxyl- and amino-termini as well as individual amino acid side chains. In our manuscript we extend this definition by changes from the canonical sequence, i.e. mutations.

To clarify these points, we have adapted the text to reflect proteolytic cleavages more generally and highlighted changes as abundance changes due to modification throughout the manuscript and included the definition of modification as used in FLEXIQuant-LF.

Changes in manuscript see our response to Essential revisions point 1.

3) Since many peptides often contain multiple potential sites of modifications (e.g. several phosphorylation sites on the same peptide), it is not possible to accurately measure the phosphorylation dynamics for each site. The authors should modify their claims and discuss this important caveat.

We thank the reviewer for the comment and hope to clarify this point in the following. We agree that it is hardly possible to accurately measure individual modification dynamics of each site when several phosphorylation events occur within the same peptide. However, we want to emphasize here that FLEXIQuant-LF, as well as the experimental version FLEXIQuant, do not to measure and identify types of modifications nor their sites but solely highlight the presence of modifications and measure the extent thereof through indirect means, i.e. by measuring the unmodified peptide. Therefore, FLEXIQuant-LF does not measure individual modification dynamics per site but does so on a peptide level. We hope the changes we made following the first comment implicitly clarify this point in the manuscript further (see Essential revisions point 1). We also added the term ‘peptide-resolved’ to modification dynamics to clarify this point further.

Changes in manuscript:

“Here, we demonstrate the utility of FLEXIQuant-LF by applying our method to interrogate the peptide-resolved modification dynamics of the anaphase-promoting complex/cyclosome (APC/C) during mitosis (Singh et al., 2009; Steen et al., 2008) ”

Changes in manuscript:

“The objective of our study was to establish a workflow to elucidate peptide-resolved modification dynamics in an unbiased manner without the need for heavy isotope-labeled reference proteins or peptides.”

Changes in manuscript:

“Prior to analysis of the peptide-resolved modification dynamics of the APC/C, the complex was purified from each time point separately by co-immunoprecipitation (co-IP) using an anti-CDC27 antibody (workflow in Figure 1A).”

Changes in manuscript:

“The peptide-resolved modification dynamics of other APC/C complex components”

Changes in manuscript:

“A more detailed analysis of the peptide-resolved modification dynamics observed in APC1 identified two regions within the protein that appear to have different modification kinetics (Figure 3). […] These data demonstrate that tools to study the peptide-resolved modification dynamics in a quantitative manner will greatly improve the current understanding of biological processes.”

Changes in manuscript:

“Furthermore, FLEXIQuant-LF can be used to repeatedly analyze the same data set with varying reference samples, providing complementary perspectives on the peptide-resolved modification dynamics.”

Changes in manuscript:

“The application of FLEXIQuant-LF to a DIA dataset of the APC/C isolated from Hela cells arrested at various time points during a mitotic arrest, allowed us to recapitulate the peptide-resolved modification dynamics previously observed with our original heavy isotope-labeled internal standard-based FLEXIQuant method. Using FLEXIQuant-LF, we easily extended our analysis to other APC/C subunits present in the sample set and delivered a comprehensive report on the peptide-resolved modification dynamics of the core components of APC/C following nocodazole-induced mitotic arrest.”

Changes in manuscript:

“Therefore, FLEXIQuant-LF can pave the way to a better general understanding of peptide-resolved post-translational modification dynamics of proteins.”

4) Although the authors state that FLEXIQuant-LF allows quantification of the modification extent without prior knowledge of the type of modification, they should explicitly state that after this analysis, the exact type of modification is still large unknown.

We thank the reviewer for the comment. Although we never claim to identify the exact type of modification, we agree that our wording in the manuscript does not explicitly state this fact. We have added clarifications in the Abstract and conclusion of the manuscript to further highlight this point.

Changes in manuscript:

“Here, we introduce FLEXIQuant-LF, a software tool for large-scale identification of differentially modified peptides and quantification of their modification extent without knowledge of the types of modifications involved.”

Changes in manuscript:

“FLEXIQuant-LF is unbiased towards the type of modification and relies on identification of the unmodified counterpart of a modified peptide to detect and quantify modification events, irrespective of the modification’s nature, i.e. the types of modifications involved remain unknown. The valuable information of modification event detection and modification extent quantification can subsequently be used to study rare PTMs and to potentially discover novel PTMs using more targeted approaches in future experiments.”

5) The authors should carry out an artificial positive control experiment using synthetic peptides to provide better understanding to readers of how well the algorithm works.

We thank the reviewer for the suggestion. The use of synthetic peptides with and without modifications is challenging to use as a positive control as FLEXIQuant-LF relies on other peptides of the protein. Spiking in synthetic peptides with modifications changes the overall abundance of this peptide compared with the remaining peptides of a protein and therefore would not be suitable as a positive control. Similarly, using synthetic proteins with modifications the purity and overall changes of modifications provides a suboptimal positive control experiment. To address the point raised by the reviewer we have added additional analyses of an independent DIA experiment with replicates by Bruderer et al. (Bruderer et al., 2017, publicly available data downloaded from the PRIDE repository, PXD005573) and changed randomly based on a uniform distribution the intensities of peptides by multiplying by a factor between 0.0 and 1.0 representing a modification extent of 100% to 0%, respectively, and tested whether FLEXIQuant-LF could recover the expected change. We assessed this both qualitatively, i.e. classifying the peptide as likely not differentially modified, possibly differentially modified, or likely differentially modified, and quantitatively, i.e. whether the RM-score difference between *in-silico* changed and unchanged peptides represents the factor. This procedure was repeated on 122 proteins with multiple peptides per protein resulting in a total of 484 peptides being tested across replicates. Classification of peptides as likely not differentially modified and likely differentially modified were identified with sensitivity and precision of >96%. Due to the hard-to-define border of the narrow group of possibly differentially modified peptides situated in the grey zone between the clean cut groups of likely and likely not differentially modified peptides results are reduced in sensitivity of 72.2% and precision of 88.6%. This decrease in accuracy likely reflects more on the definition criteria (boundaries of RM-score > 0.5 and RM-score < 0.6) than on the method itself. Additionally, the RM score also shows a high correlation with the expected in-silico modification changes (Pearson’s r=0.98). We have added two panels to Figure 4 highlighting the results and added a paragraph in the Materials and methods and Results section describing the additional analyses.

Changes in manuscript:

“Reproducibility and quality assessment

[...] To further test the quality of the classification and quantification accuracy an independent publicly available DIA dataset was downloaded from the PRIDE repository (PXD005573) (Bruderer et al., 2015). […] Furthermore, total RM score errors as defined by the difference between the expected change factor and the calculated RM score changes were compared to the number of peptides measured per protein to assess robustness of the FLEXIQuant-LF approach.”

Changes in manuscript:

“Testing and Validating Reproducibility of FLEXIQuant-LF

[…] We also tested the quality of FLEXIQuant-LF classification and quantification on an independent DIA dataset (Bruderer et al., 2015). [...] RM score changes showed a high correlation with the in-silico intensity change factors (Pearson's r=0.98, Figure 4C). [...] Overall, quantification errors were low with a mean error of 0.020 (SD=0.058, <0.1 for 94.7% of cases, Figure 4E). […] Outliers are rare (<5%) and can be attributed to the random selection and simulation effects of the assessment approach.”

Changes in manuscript:

“Figure 4. FLEXIQuant-LF reproducibility and error estimation. […] Overall quantification errors are very low and further improve with the number of peptides used for FLEXIQuant-LF analysis.”

6) The authors should apply FLEXIQuant-LF to one of publicly available label-free phosphoproteome datasets with known answers regarding the extent of phosphorylation? If the authors can do this and obtain similar conclusions, then it can indeed be presented as a powerful strategy that can be applied in many studies.

We thank the reviewer for the comment. FLEXIQuant-LF is a label-free based method that identifies and quantifies the modification extent of peptides within a protein in an indirect manner by measuring the abundance change in unmodified peptides. By design this method is agnostic of the modification type and stoichiometry of different modification sites within a peptide. As the reviewer had pointed out previously, we have clarified these aspects of our strategy (see Essential revisions point 1). Unfortunately, label-free phosphoproteome data, which are based on phosphopeptide-enrichment, do not provide the basis on which FLEXIQuant-LF operates. Even combining label-free global data and label-free phosphoproteome data suffer from the presence of additional modifications that are inherently included in the agnostic FLEXIQuant-LF method but are neglected by enrichment methodologies. We appreciate the reviewer’s comment to boost the power of our strategy but have to highlight, that no published dataset is available to date to follow the suggestion by the reviewer as too many confounding factors, i.e. additional modifications, enrichment protocols for specific modifications only, exist.